# Metabolomic Analysis of Respiratory Epithelial Lining Fluid in Patients with Chronic Obstructive Pulmonary Disease—A Systematic Review

**DOI:** 10.3390/cells12060833

**Published:** 2023-03-08

**Authors:** Kaja Pulik, Katarzyna Mycroft, Piotr Korczyński, Andrzej K. Ciechanowicz, Katarzyna Górska

**Affiliations:** 1Department of Internal Medicine, Pulmonary Diseases and Allergy, Medical University of Warsaw, Banacha 1A, 02-097 Warsaw, Poland; 2Laboratory of Regenerative Medicine, Medical University of Warsaw, Banacha 1B, 02-097 Warsaw, Poland

**Keywords:** BALF, cigarette smoke, COPD, EBC, induced sputum, lipidomics, metabolic alterations, metabolome, metabolomics

## Abstract

Chronic obstructive pulmonary disease (COPD), as the third leading cause of death among adults, is a significant public health problem around the world. However, about 75% of smokers do not develop the disease despite the severe smoking burden. COPD is a heterogeneous disease, and several phenotypes, with differences in their clinical picture and response to treatment, have been distinguished. Metabolomic studies provide information on metabolic pathways, and therefore are a promising tool for understanding disease etiopathogenesis and the development of effective causal treatment. The aim of this systematic review was to analyze the metabolome of the respiratory epithelial lining fluid of patients with COPD, compared to healthy volunteers, refractory smokers, and subjects with other lung diseases. We included observational human studies. Sphingolipids, phosphatidylethanolamines, and sphingomyelins distinguished COPD from non-smokers; volatile organic compounds, lipids, and amino acids distinguished COPD from smokers without the disease. Five volatile organic compounds were correlated with eosinophilia and four were associated with a phenotype with frequent exacerbations. Fatty acids and ornithine metabolism were correlated with the severity of COPD. Metabolomics, by searching for biomarkers and distinguishing metabolic pathways, can allow us to understand the pathophysiology of COPD and the development of its phenotypes.

## 1. Introduction

### 1.1. Rationale

Chronic obstructive pulmonary disease (COPD) is a heterogeneous group of lung diseases characterized by persistent obstruction of the airways and progressive decline in lung function caused by chronic bronchitis and/or emphysema. In 2019, about 10% of people aged 30 to 79 were affected, corresponding to almost 0.4 billion people worldwide [1]. The main risk factor for COPD is long-term exposure to tobacco smoke, which, at the cellular level, involves the response of epithelial cells, including oxidative stress, subclinical inflammation, cell apoptosis, cell remodeling, and cell dysfunction [2].

Approximately one-quarter of smokers have COPD [3]. However, to this day, it is not clear why some smokers develop COPD and others do not. Several hypotheses have been proposed, including poor lung development during the fetal period and childhood or concurrent asthma leading to the development of COPD [4]. It is also unclear why COPD manifests itself through clinical phenotypes that differ in clinical course, the incidence of comorbidities, approach to therapy, and prognosis. Known phenotypes include the predominance of emphysema, chronic bronchitis, frequent exacerbations, and the co-occurrence of COPD with asthma. Appendix A showing the phenotypes of COPD, their diagnosis, prognosis, and phenotype-specific treatment is included in the supplementary material [1,5,6,7].

Lack of knowledge of the etiopathogenesis of the different manifestations of the disease results in ineffective treatment based on the historical approach that ‘one size fits all’. Therefore, new state-of-the-art tools (e.g., high-resolution mass spectrometry) that allow biomarker discovery at a more profound level are currently used. Metabolomics is a field of systems biology that focusses on qualitative and quantitative analysis of low-mass molecules (<1 kDa) that are components of not only host metabolic pathways, but also originate from symbiotic and pathogenic microorganisms, the non-organic environment, and interactions between them. The great chemical diversity of the metabolome results in the use of many analytical techniques or a combination of them (e.g., liquid chromatography coupled to mass spectrometry). Metabolomic analysis might allow the identification of points of causal treatment [8]. Additionally, it can help to find new disease phenotypes by identifying altered metabolic and signaling pathways and/or biomarkers.

Changes in metabolic profiles in plasma and serum were associated with differences in pulmonary function tests, exacerbations, and extent of emphysema [9]. In contrast, very few studies have been conducted on respiratory epithelial lining fluid (ELF). Respiratory ELF is the fluid produced by the epithelium lining the lower respiratory tract. There are various materials recognized as representing the ELF, such as exhaled breath condensate, induced sputum, or bronchoalveolar lavage fluid. Pathological changes in the signaling pathways involved in metabolic pathogenetic processes could be identified in ELF, and it is important to assess metabolic changes in situ. Notably, different airway materials correspond to particular levels of the respiratory tract [10], and various molecules could be measured.

### 1.2. Objectives

We present a systematic review to evaluate the respiratory metabolome in patients with COPD with respect to clinical characteristics, functional tests, and imaging studies. Furthermore, our objective was to assess whether the respiratory metabolome allows differentiation between clinical phenotypes of COPD.

## 2. Materials and Methods

### 2.1. Protocol and Registration

The protocol web address published on the PROSPERO website https://www.crd.york.ac.uk/PROSPERO/display_record.php?RecordID=242510 accessed on 12 April 2021 with assigned registration number: CRD42021242510. The systematic review was conducted following the Preferred Reporting Items for Systematic Reviews and Meta-Analyses (PRISMA) guidelines.

### 2.2. Eligibility Criteria

We considered observational studies involving adults with COPD defined by GOLD criteria or national recommendations, smokers without a diagnosis of the disease, and non-smokers. We include studies carried out on materials obtained from the airways. We evaluated differences in metabolite concentrations measured in ELF collected by non-invasive methods: exhaled breath (EB), exhaled breath condensate (EBC), induced sputum (IS), and invasive methods: bronchoalveolar lavage fluid (BALF) and bronchial biopsy material. Acceptable control groups were non-smokers, smokers, and patients with diseases other than COPD. Publications investigating the impact of including steroid treatment were rejected to avoid the effect of glucocorticosteroids on the metabolome. Cell-line or animal studies were not considered. Studies with fewer than 10 participants in a single group were not included. Studies with published status were considered.

### 2.3. Information Sources

We searched the following electronic bibliographic databases: EMBASE, PubMed, and Web of Science. The search started on 3 March 2021; the last search took place on 3 October 2021.

### 2.4. Search

The search strategy only included terms related to metabolomics. Keyword combinations were ‘chronic obstructive pulmonary disease’ or ‘COPD’ or ‘emphysema’ or ‘chronic bronchitis’ and ‘omics’ or ‘metabolomics’ or ‘metabonomics’ or ‘lipidomics’ or ‘metabolic pathway’. We went through the citations to look for relevant studies that we could include in the review.

### 2.5. Study Selection

Two independent investigators reviewed the abstracts of articles and the full texts if the abstract did not provide the information necessary to include or exclude the study. In the case of doubt, a third researcher was asked for her opinion. We excluded non-original articles, reviews, letters to the editor, comments, and case reports. Due to insufficient information and lack of reviews of retrieval validity, we excluded conference reports and articles where only the abstract was available. All articles that met the criteria were published in English. The repetitions were removed automatically; then, the remaining repetitions were manually checked and removed.

### 2.6. Data Items

Studies on the metabolome that included adults with COPD were sought. Studies without a control group were excluded. All animal and cell line studies were excluded. Studies on systemic materials such as plasma, serum, and urine were excluded. Studies that included subjects with alpha-1-AT deficiency or asthma—COPD overlap were excluded.

### 2.7. Risk of Bias

The risk of error was determined by assessing the inclusion of PICO, the presence of a control group, and determining the minimum group size in the articles included in the review.

## 3. Results

### 3.1. Study Selection

We found 216 articles in the PubMed database, 279 articles in the Web of Science database, and 295 articles in EMBASE, resulting in 790 articles. An automatic search for repetitions in Endnote revealed the presence of 252 repetitions, which we excluded from the review. We then manually reviewed the titles for repetitions, finding an additional 130. After reviewing the abstracts and, if necessary, due to lack of complete information in the abstract, the full text, 386 articles were rejected (Figure 1). Of the 22 studies included in the thorough assessment, eight were rejected. In two articles, participants with COPD caused by alpha-1-AT deficiency constituted the research group; in the following two studies, the study groups were subjects with co-existing asthma and COPD; in one study, we could not obtain the full text; 3 studies were rejected due to lack of a control group. Finally, we include 14 studies in the review.

### 3.2. Results

We found 14 studies on metabolomics analysis in ELF. The detailed characteristics of each study, the biological materials used, and the biomarkers evaluated are shown in Table 1, Table 2, Table 3 and Table 4. Most authors used the non-invasive measurements of EBC (in one study, additional material, saliva, was used); some used real-time measurements of exhaled breath. A less common non-invasive material was IS. Two studies performed an invasive acquisition of BALF during bronchoscopy. In most studies the control group consisted of smokers without COPD and/or non-smokers. All these studies have shown significant differences in the concentration of metabolites in all ELF materials, real-time analysis of exhaled breath [11,12,13], EBC [14,15,16,17], IS [18], and BALF [19] between never-smokers, smoking controls, and patients with COPD. Figure 2 presents a graphical summary of the results.

#### 3.2.1. Exhaled Breath

Four studies of exhaled breath were performed; two using real-time eNose measurements [13,23] and two using mass spectrometry [11,12]. Regardless of the method used, studies have confirmed that the specificity and sensitivity of EB-spectra-based profiles of volatile organic compounds (VOCs) are sufficient to distinguish between COPD and non-smokers [11,12,13]. The only study on metabolomics in EB that included smoking controls was published in 2014 by Sinues et al. [12]. The model distinguished COPD from smoking controls with 96% sensitivity and 72.7% specificity. Positively correlating with COPD biomarkers were VOCs such as alpha-pyrene, acetaldehyde, 2-butyloctanol, octane, methyl isobutyrate, butanal, 2-propanol, 3-hexanone, cyclopentanone, 3-methyl-propanal, whereas the negative predictors were delta-dodecalactone, 2-methyl butanoic acid, 2-acetylpyridine, tetradecane, cinnamaldehyde, vinylpyrazine [13] and undecanal [11]. The discriminatory ability of the EB-based model was also high for the severity of airflow limitation; it differentiated GOLD I/II from GOLD III/IV with a sensitivity of 92.3% and a specificity of 83.3%. In another study, 17 VOCs were correlated with the GOLD 2019 stages [13]. Furthermore, exhaled breath analysis allowed the authors to identify a phenotype with ≥2% sputum eosinophilia and frequent exacerbations (ROC area under curve, 0.94 and 0.95, respectively) [11]. EB metabolomics also allowed for a distinction between asthma and COPD. Fens et al. (2011) included patients with asthma with airway obstruction (reversible and irreversible; *n* = 39 and 21, respectively) and patients with COPD (GOLD stages II-III; *n* = 40) [23]. EB metabolomic fingerprints provided an excellent tool for distinguishing asthma with fixed obstruction from patients with COPD with a sensitivity of 85% and a specificity of 90%; results did not depend on the current smoking status.

#### 3.2.2. Exhaled Breath Condensate

Five of the six EBC studies were conducted using nuclear magnetic resonance [14,15,16,24,25], and one was conducted using mass spectrometry [17]. The key particles differentiating COPD from non-smokers and smokers in the NMR-based studies were VOCs, while in the MS-based study, sphingomyelins were the differentiating particles. Similar to EB, EBC metabolomics also significantly differentiated non-smokers and smokers from COPD patients [14,15,16,17]. With COPD versus non-smokers, there was a positive correlation of compounds of the glycolysis pathway or compounds derived from bacteria (lactate, acetate, propionate) and amino acids (proline, serine, tyrosine) [16]. Similarly, there was a positive correlation of acetate with COPD versus smokers [15]. Negatively correlated particles with COPD versus non-smokers were acetone, valine, and lysine [16], and in COPD versus smokers, it was l-methylimidazole [15]. COPD compared to non-smokers had lower sphingomyelin levels, a more varied phosphatidylcholine profile, and higher accumulation of lysophosphatidylcholines [17]. EBC spectra were different between COPD phenotypes varying in the extent of emphysema [16]. In another study, the stages of COPD, according to GOLD 2017, differed in the expression of unsaturated fatty acids, ornithine metabolism compounds, and plasma protein signals [17].

EBC-based studies were able to distinguish COPD from other diseases. In 2018, Maniscalco et al. conducted a study that included EBC of patients with asthma (*n* = 20) and COPD (*n* = 32) [24]. Metabolomic analysis distinguished the two study groups with a regression of 95% and *p* < 0.0013. An increase in ethanol and methanol concentrations and a decrease in formate and acetone were observed in the COPD group, compared to asthmatic patients. In the study published in 2013 by de Laurentiis et al. to assess the discriminatory ability of EBC metabolomic fingerprinting in tobacco-related diseases, EBC was collected from smoking controls (*n* = 20), patients with COPD (*n* = 15), and histiocytosis (*n* = 15) [15]. The EBC fingerprints distinguished the three groups with 96% accuracy; they obtained R2 = 0.97 and Q2 = 0.91 for smoking controls versus COPD, R2 = 0.87 and Q2 = 0.79 for histiocytosis versus smoking controls, and R2 = 0.90 and Q2 = 0.81 for COPD versus histiocytosis. In COPD, in contrast to histiocytosis, increased levels of 2-propanol and decreased levels of isobutyrate were observed. In 2008, the authors published a study in which they included never-smokers (*n* = 12), patients with COPD (*n* = 12), and laryngectomized patients due to laryngeal cancer (*n* = 12); the procedure was performed at least one year before collection [14]. The NMR spectra between saliva and the corresponding EBC samples differed for each subject. Both the EBC and saliva spectra also differed significantly between the study groups, but specific metabolites were not provided. The study by Ząbek et al. in 2015 included 18 subjects with COPD and 28 subjects with obstructive sleep apnea (OSA) [25]. The EBC did not show a good ability to discriminate between the two groups.

#### 3.2.3. Induced Sputum

The first study on the metabolome in IS was published by Telenga et al. in 2014 [18]. They included never-smokers (*n* = 14), smoking controls (*n* = 20), and COPD patients (GOLD stage II/III; *n* = 19). More than 1500 lipids were identified. A higher expression of sphingolipids was observed in the COPD group compared to smoking controls, and a higher expression of glycosphingolipids in smoking controls compared to never-smokers. Moreover, this is the only study that included follow-up. In smokers, measurements were taken both during smoking and two months after smoking cessation. Compared to baseline, down-regulation of 26 sphingolipids was observed in both study groups two months after cessation of smoking. These results indicate that smoking affects metabolite concentrations, but, more importantly, some of these changes are reversible after stopping smoking. The study by t’Kindt et al. in 2015 included 17 patients with COPD, 20 smoking controls, and 14 never-smokers [20]. This study was conducted on the same cohort as the Telenga et al. study; the analysis of IS compounds confirmed that smokers without and with COPD differ significantly in sphingolipid expression.

#### 3.2.4. Bronchoalveolar Lavage Fluid

Walmsley et al. (2018) created a BALF metabolomic database derived from samples from 117 individuals, of which 13 were non-smokers, 77 were smokers without a diagnosis of COPD, and 47 had a diagnosis of COPD [19]. Analysis revealed the presence of more than 11,000 lipids and approximately 650 water-soluble substances in BALF. One-tenth of these substances were found in all samples. Halper-Stromberg et al. conducted a highly relevant study in 2019 [22]. The study group consisted of COPD patients (*n* = 47), smoking controls (*n* = 56), and never-smokers (*n* = 12). In total, 115 BALF samples were collected, in which approximately 8000 unique molecules were detected. The BALF particle count was correlated with FEV_1_/FVC (1230 particles), emphysema (almost 800 particles), FEV_1_% predicted (8 particles), and the frequency of COPD exacerbations (1 particle).

## 4. Discussion

To our knowledge, this is the first systematic review of the literature on ELF metabolomics in patients with COPD. In total, 14 studies were included that compared the airway metabolome between patients with and without COPD. The compounds that differentiated patients with COPD and those without COPD belong to classes of lipids, VOCs, and amino acids. However, due to differences in airway materials representing ELF and measurement methods of the metabolites, the data are heterogeneous and the results were not replicated by several authors. Furthermore, very few reports on respiratory metabolomics of various COPD phenotypes are available. Although COPD is considered a systemic disease, its early phase begins in the airways that are exposed to tobacco smoke and air pollution [26]. Therefore, to investigate the role of altered airway metabolome in the COPD pathomechanism, we focused on materials representing airway lining fluid.

The altered airway lipidome in COPD patients has been found in several studies that compared COPD to both smoking and never-smoking non-COPD patients [17,18,19,20]. Higher expression of bioactive molecules, such as sphingolipids, phosphatidylethanolamines, and sphingomyelins, was shown in COPD subjects compared to non-COPD subjects, both in IS [18,20] and EBC [17]. However, these changes in the lipidome are present not only in the airways, but also in serum [26,27]. Dysregulation of lipid pathways may be an essential pathomechanism in the development of COPD. Smoking cigarettes raises the level of sphingolipids, which increases oxidative stress and promotes apoptosis, thus contributing to the development of chronic inflammation in the airways [28]. In a different study, a two-month smoking cessation period resulted in a decrease of 26 sphingolipids in the smoking controls and COPD groups compared to baseline [18]. It has been hypothesized that hypolipidemic treatment might improve COPD prognosis. The study by Lu et al. showed a reduction in patients with all-cause mortality in COPD after statin inclusion [29]. There have also been attempts to investigate other substances in the treatment of COPD, including inhibitors of enzymes in sphingolipid pathways such as sphingosine kinase, S1P lyase, acid sphingomyelinase, and sphingosine-1-p receptor agonists, but the results are not conclusive [30,31,32]. Finding differences in lipid metabolism between COPD and smokers can reveal new directions for therapy.

Another group of substances in ELF, which differed between COPD and non-COPD smokers or never-smokers, were VOCs and amino acids. Most VOCs (e.g., acetate, propionate, acetaldehyde, alpha-pyrene, 2-butyloctanol) and amino acids (e.g., serine, proline tyrosine) that distinguish COPD from non-COPD subjects participate in oxidative stress reactions. This mechanism, along with an increase in alveolar apoptosis, is considered the main pathomechanism for the development of emphysema [33]. Increased proline is associated with response to environmental stressors and is upregulated in EBC in COPD patients [16]. Elevated proline levels in COPD patients versus non-smokers were also recently observed in a study on induced sputum conducted on a large group of patients. (Esther 2022) [21].

Some substances in ELF might have bacterial origin [16,34]. An increase in lactate has been proven in exacerbations of cystic fibrosis. Effective antibiotic therapy decreased lactate concentrations in ELF [34]. In turn, higher lactate concentrations in stable COPD indicate the involvement of subclinical bacterial bronchitis in the pathogenesis of COPD.

The differences in airway metabolome have also been found in different COPD phenotypes. The search for biomarkers or metabolic pathways that distinguish the different phenotypes of COPD is crucial for understanding their distinct pathomechanisms, but also for the search for early diagnosis, personalized treatment, and prognostic indicators. Differences in airway metabolome specific to certain disease phenotypes have been demonstrated in three studies included in the review [11,16,22] and one study which was published after search completion [21]. The EBC spectra could distinguish clusters with and without emphysema dominance [16]. In BALF, 800 particles were associated with aggravation of emphysema, mainly phospholipids, lysophospholipids, and essential amino acids; one molecule distinguished the phenotype with frequent exacerbations [22]. A study conducted on induced sputum on 562 COPD patients showed a significant elevation of concentrations of sialic acid, hypoxanthine, xanthine, methylthioadenosine, adenine, and glutathione in patients with the phenotype with chronic bronchitis compared to patients without chronic bronchitis [21]. In EB, four VOCs were related to the number of disease exacerbations during the year preceding the study, and five VOCs were associated with eosinophilia in the sputum [11]. Furthermore, this metabolomic analysis distinguished COPD patients receiving and not receiving inhaled steroids. This correlation may indicate that the compounds that distinguished the phenotype with higher eosinophilia might not be related to the disease pathogenesis itself, but to the more frequent use of inhaled steroids in this group of patients.

The metabolite particles in the airways appear to have a prognostic value, which was found in two studies published after the search period [21,35]. In a study on IS involving a group of more than 500 COPD patients, an association of elevated sialic acid and hypoxanthine with a shorter time to exacerbation was demonstrated [21]. The addition of these two parameters to a clinical model for the prediction of exacerbations (history of previous exacerbations, percent predicted FEV1, smoking, and age) significantly improved the sensitivity of the test. This finding could be significant for clinicians in predicting COPD exacerbations, detailing the particular group at risk. Furthermore, sialic acid has a direct association with excessive mucus production, highlighting the involvement of this pathomechanism in COPD exacerbations. Increased mucus hydration in the airways may translate into a lower risk of exacerbations, and lowering sialic acid appears to be a potentially useful biomarker of airway mucus hydration. Research on patients with COPD undergoing lung rehabilitation revealed that improvements in clinical symptoms were correlated with changes in the EBC metabolome [35]. The decrease in dyspnea and the increase in distance in the 6 min walk test were associated with a reduction in methanol concentrations. This is the first study to demonstrate the clinical effectiveness of biomarkers, and the authors are working on a tool to measure methanol in the EBC in real-time. They emphasize the superiority of objective measurement of the effectiveness of rehabilitation over subjective tools dependent on patients.

The search for molecules linked to clinical results might identify objective markers and predict factors for a worse course of the disease. A model based on differences in EB spectra distinguished the severity of airflow limitation [12]. An association of four phospholipids and sphingomyelin with FEV_1_/FVC was observed [22]. Pulmonary surfactant mainly consists of phospholipids, and its properties depend on the percentage of individual molecules; for example, the higher the concentration of phosphatidylcholine 16:0/16:0, the higher the resistance to high pressures. In patients with COPD, a decrease in the proportion of phospholipids in the surfactant building may be associated with reduced lung function [26]. The severity of the disease varied with fatty acid concentrations and ornithine metabolism in EBC [17] and 17 VOCs in EB [13]. Systemic metabolomics based on lipids presented similar results. In 2019, Yu et al. published a systematic review on systemic biomarkers in COPD where the authors identified dozens of systemic metabolites associated with lung functional outcomes, mainly with FEV_1_ and FVC, and less with FEV_1_/FVC [36]. The authors explained this by the greater effect of lung size reduction on metabolomics than bronchoconstriction alone.

### Limitations

This review has several limitations. Firstly, most of the studies have small sample sizes, indicating the need for more extensive studies. Secondly, due to the heterogeneity of the designs of studies, meta-analysis was not possible. The studies differed in the inclusion and exclusion criteria of the study and control groups. The COPD patients varied in the severity of the disease, and the control groups were never-smokers, non-COPD smokers, and patients with other lung diseases. In addition, heterogeneity is also applied to methods and materials, making comparison difficult. From the clinical point of view, a few studies took into account symptoms of the disease such as dyspnea or fatigue or phenotypes. In addition, only two of them measured metabolites over time. In order to investigate the usefulness of metabolites as biomarkers or predictors, studies with metabolome measurements over time are urgently needed, taking into account changes in clinical parameters, including clinical symptoms. Figure 3 outlines future research directions in the field of metabolomics in COPD and its phenotypes [37].

## 5. Conclusions

Metabolomic analysis of ELF is a promising approach to understanding the pathomechanisms of COPD development and its clinical phenotypes, such as emphysema, eosinophilic phenotype and frequent exacerbations. Existing studies, conducted on small groups, indicate that metabolomic measurements can be used to distinguish COPD patients from both non-smokers and smokers without COPD. The patients also differ in their metabolic profiles due to their lung function scores and respiratory imaging findings, such as the extent of emphysema. In addition, some studies indicate that COPD can be distinguished from other lung diseases such as asthma, even with fixed bronchial obstruction, with high sensitivity and specificity. The area of metabolomics in ELF is attracting growing attention because of the possibility of understanding the pathogenesis of diseases on a more profound, metabolic level.

The abovementioned differences could facilitate the identification of individuals susceptible to developing the disease before the progression of irreversible changes in the lung parenchyma occurs. In addition, the identification of specific metabolomics changes can lead to the development of new therapeutic strategies.

On the other hand, the low reproducibility of metabolites between studies despite their promising results, as observed in this review, is the subject of current debate. In the metabolome of COPD, particularly in the respiratory metabolome, we are at the stage of case-control studies, and thus the studies presented here have an exploratory character. It is postulated that results of case-control studies in metabolomics, especially on small groups, may not be reflected in studies on larger cohorts due to the huge amount of data coming from the measurement of thousands of molecules, the influence of lifestyle, and the large intra-individual and inter-individual variability.

Furthermore, we are aware that none of the included papers was designed to compare phenotypes. Therefore, we have attempted to extract information on this topic from the included studies, which we presented in Table 3.

Further studies on ELF metabolic biomarkers of various COPD phenotypes are needed. Studies measuring the metabolome in a group of smokers before the development of the disease and at a different time point could be a valuable source of information on biomarkers that are predictors of the development of the disease.

## Figures and Tables

**Figure 1 cells-12-00833-f001:**
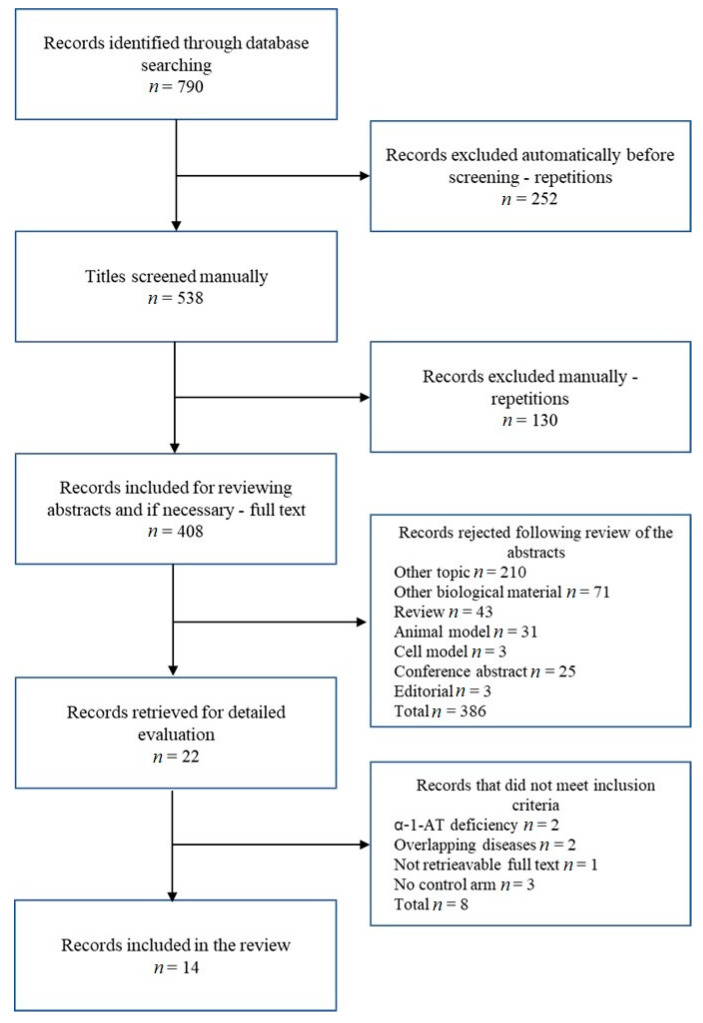
Diagram showing the number of studies and reason for excluding studies.

**Figure 2 cells-12-00833-f002:**
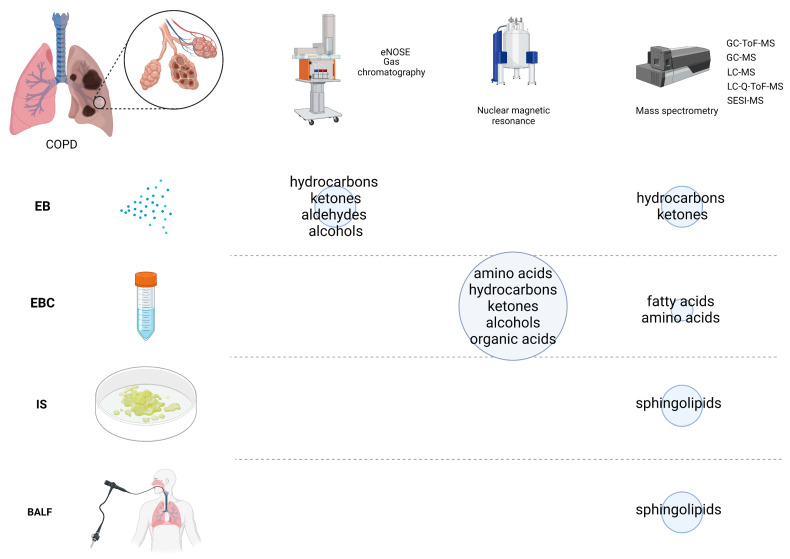
Groups of molecules that significantly differentiated COPD are shown. The size of the blue circles corresponds to the number of studies in a given material and method. Created with BioRender.com.

**Figure 3 cells-12-00833-f003:**
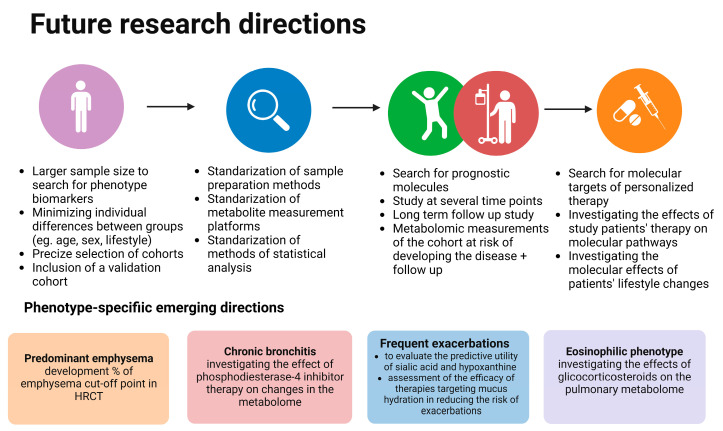
Future research directions in the field of COPD metabolomics. Created with Biorender.com.

**Table 1 cells-12-00833-t001:** Comparison of chronic obstructive pulmonary disease patients and non-smokers.

Author, Year	Population	Material	Method	Key Differences
de Laurentiis et al., 2008 [14]	COPD *n* = 12 64.9 ± 5.7 years never-smokers *n* = 12 55.6 ± 7.2 years	EBC	NMR	The EBC spectra differed between COPD and controls.
Basanta et al., 2012 [11]	COPD *n* = 39 65.7 ± 6.8 years never-smokers *n* = 32 55.3 ± 7.1 years	EB	GC-ToF MS	Positive → Undecanal
Bertini et al., 2014 [16]	COPD *n* = 37 70 (66–74) years never or ex-smokers *n* = 2556 (54–64) years	EBC	NMR	Positive → Lactate → Acetate → Propionate → Serine → Proline → Tyrosine Negative → Acetone → Valine → Lysine
Kilk et al., 2018 [17]	COPD *n* = 25 67 (58–72) years never-smokers *n* = 21 37 (27–62) years	EBC	LC MS	Negative → Sphingomyelins
Sinues et al., 2014 [12]	COPD GOLD I/II *n* = 13 63 ± 7 years GOLD III/IV *n* = 12 62 ± 5 years never-smokers *n* = 25 27 ± 9 years	EB	SESI MS	COPD and never-smokers differed by a metabolic panel (96% sensitivity, 72.7% specificity).
Rodriguez-Aguilar et al., 2019 [13]	COPD *n* = 23 67.7 ± 8.6 years ex- and never-smokers *n* = 33 55.6 ± 8.4 years	EB	FGC eNose	Positive → Alpha-pyrene → Acetaldehyde → 2-butyloctanol → Octane → Methyl isobutyrate → Butanal → 2-propanol → 3-hexanone → Cyclopentanone → 3-methyl-propanal Negative → Delta-dodecalactone → 2-methyl butanoic acid → 2-acetylpyridine → Tetradecane → Cinnamaldehyde → Vinylpyrazine
t’Kindt et al., 2015 [20]	COPD *n* = 17 59 (54–65) years never-smokers *n* = 14 54 (23–58) years	IS	LC-Q-ToF MS	Positive → Sphingolipids
Esther et al., 2022 [21]	COPD GOLD I *n* = 178 66.4 ± 8.4 years GOLD II *n* = 303 64.9 ± 7.8 years GOLD III *n* = 81 64.8 ± 8.3 years never-smokers *n* = 77 55.4 ± 10.2 years	IS	LC-MS	Positive → Sialic acid → Hypoxantine → Xantine → Methylthioadenosine → Adenine → Glutathione
Walmsley et al., 2018 [19]	COPD *n* = 47 never-smokers *n* = 13 age not specified	BALF	LC-Q-ToF MS	Higher number of lipid compounds in smoking controls compared to never-smokers or ex-smokers

Abbreviations: BALF—bronchoalveolar lavage fluid; COPD—chronic obstructive pulmonary disease; EB—exhaled breath; EBC—exhaled breath condensate; FGC—fast gas chromatography; GC-ToF MS—gas chromatography–time-of-flight mass spectrometry; IS—induced sputum; LC—liquid chromatography; LC MS—liquid chromatography mass spectrometry; LC-Q-ToF MS—liquid chromatography quadrupole time-of-flight mass spectrometry; NMR—nuclear magnetic resonance; SESI MS—secondary electro-spray ionization mass spectrometry.

**Table 2 cells-12-00833-t002:** Comparison of chronic obstructive pulmonary disease patients and smokers.

Author, Year	Population	Material	Method	Key Differences
de Laurentiis et al., 2013 [15]	COPD *n* = 15 66.9 ± 9.9 years smoking controls *n* = 20 41.9 ± 12.9 years	EBC	NMR	Positive → Acetate Negative → 1-methylimidazole
Sinues et al., 2014 [12]	COPD GOLD I/II *n* = 13 63 ± 7 years GOLD III/IV *n* = 12 62 ± 5 years smoking controls *n* = 11 34 ± 10 years	EB	SESI MS	Positive → Acetone Negative → Indole
Telenga et al., 2014 [18]	COPD *n* = 19 59 (54–65) years smoking controls *n* = 20 42 (20–51) years	IS	LC-Q-ToF MS	Positive → 168 sphingolipids → 36 phosphatidylethanolamines After two months of smoking cessation → a decrease of 26 sphingolipids in smokers with and without COPD.
t’Kindt et al., 2015 [20]	COPD *n* = 19 59 (54–65) years smoking controls *n* = 20 42 (20–51) years	IS	LC-Q-ToF MS	Positive → Sphingolipids
Walmsley et al., 2018 [19]	COPD *n* = 47 smoking controls *n* = 77 age not specified	BALF	LC-Q-ToF MS	Higher number of lipid compounds in smoking controls compared to never-smokers or ex-smokers.

Abbreviations: BALF—bronchoalveolar lavage fluid; COPD—chronic obstructive pulmonary disease; EB—exhaled breath; EBC—exhaled breath condensate; IS—induced sputum; LC—liquid chromatography; LC-Q-ToF MS—liquid chromatography quadrupole time-of-flight mass spectrometry; NMR—nuclear magnetic resonance; SESI MS—secondary electro-spray ionization mass spectrometry.

**Table 3 cells-12-00833-t003:** Comparison of clinical outcomes and phenotypes.

Author, Year	Population	Material	Method	Key Differences
		FEV_1_/FVC		
Halper-Stromberg et al., 2019 [22]	COPD *n* = 47 64 (58–68) years smoking controls *n* = 56 58 (50–66) years never-smokers *n* = 12 56 (50–60) years	BALF	LC MS	Positive: → 4 phosphatidylethanolamines → 4 phosphatidylcholines → 2 cardiolipins → Homocysteine → 1 sphingolipid → 1 sphingomyelin → 2 glycerolipids Negative → Ceramide (d18:1/16:0)
		COPD severity		
Sinues et al., 2014 [12]	COPD GOLD I/II *n* = 13 63 ± 7 years GOLD III/IV *n* = 12 62 ± 5 years	EB	SESI MS	The metabolic panel distinguished between mild and severe COPD.
Kilk et al., 2018 [17]	COPD *n* = 25 67 (58–72) years never-smokers *n* = 21 37 (27–62) years	EBC	LC MS	Unsaturated fatty acids and ornithine metabolism differed between GOLD categories.
Esther et al., 2022 [21]	COPD GOLD I *n* = 178 66.4 ± 8.4 years GOLD II *n* = 303 64.9 ± 7.8 years GOLD III *n* = 81 64.8 ± 8.3 years	IS	LC-MS	Positive → Sialic acid → Sialic acid to urea ratio → Hypoxantine → Xantine
		Eosinophilia		
Basanta et al., 2012 [11]	COPD *n* = 39 65.7 ± 6.8 years never-smokers *n* = 32 55.3 ± 7.1 years	EB	GC-ToF MS	→ α-methylstyrene → Cyclohexenol → Benzofuran → Decane → Biphenyl
		Exacerbations		
Basanta et al., 2012 [11]	COPD *n* = 39 65.7 ± 6.8 years never-smokers *n* = 32 55.3 ± 7.1 years	EB	GC-ToF MS	→ Undecane → Tetramethyloctane → Methanoazulene → Naphthalene
Esther et al., 2022 [21]	COPD GOLD I *n* = 178 66.4 ± 8.4 years GOLD II *n* = 303 64.9 ± 7.8 years GOLD III *n* = 81 64.8 ± 8.3 years	IS	LC-MS	→ Sialic acid → Hypoxantine
		Emphysema		
Halper-Stromberg et al., 2019 [22]	COPD *n* = 47 64 (58–68) years smoking controls *n* = 56 58 (50–66) years never-smokers *n* = 12 56 (50–60) years	BALF	LC MS	→ Leucine → Lysine

Abbreviations: BALF—bronchoalveolar lavage fluid; COPD—chronic obstructive pulmonary disease; EB—exhaled breath; FEV_1_—forced expiratory volume in 1 s; FVC—forced vital capacity; GC-ToF MS—gas chromatography–time-of-flight mass spectrometry; GOLD—Global Initiative for Chronic Obstructive Lung Disease; IS—induced sputum; LC MS—liquid chromatography mass spectrometry; SESI MS—secondary electro-spray ionization mass spectrometry.

**Table 4 cells-12-00833-t004:** Comparison of chronic obstructive pulmonary disease patients and patients with other diseases.

Author, Year	Population	Material	Method	Key Differences
Fens et al., 2011 [23]	COPD (stage II-III) *n* = 40 63 (49–87) years fixed asthma * *n* = 21 64 (43–76) years classic asthma * *n* = 39 35 (18–68) years	EB	FGC, e-Nose	EBC fingerprints differed between asthma with persistent obstruction and COPD (85% sensitivity, 90% specificity) and between classical asthma and COPD (91% sensitivity, 90% specificity).
Maniscalco et al., 2018 [24]	COPD *n* = 32 55.8 ± 6.2 years asthma *n* = 20 41.8 ± 6.7 years	EBC	NMR	Positive → Ethanol → Methanol Negative → Formate → Acetone
de Laurentiis et al., 2013 [15]	COPD *n* = 15 66.9 ± 9.9 years PLCH *n* = 15 34.2 ± 7.5 years	EBC	NMR	Positive → 2-propanol, Negative → Isobutyrate
Ząbek et al., 2015 [25]	COPD *n* = 18 64/(49–81) years OSA *n* = 28 54/(27–65) years	EBC	NMR	EBC fingerprint did not differentiate between patients with COPD and OSA

Abbreviations: COPD—chronic obstructive pulmonary disease; EB—exhaled breath; EBC—exhaled breath condensate; FGC—fast gas chromatography; NMR—nuclear magnetic resonance; OSA—obstructive sleep apnea; PLCH—pulmonary Langerhans’ cell histiocytosis. * fixed asthma—asthma with fixed airways obstruction, classic asthma—asthma with reversible airways obstruction.

## Data Availability

No new data were created or analyzed in this study. Data sharing is not applicable to this article.

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
