# Peer review of "Metabolomic Analysis of Respiratory Epithelial Lining Fluid in Patients with Chronic Obstructive Pulmonary Disease—A Systematic Review"

_cells, 2023, doi:10.3390/cells12060833_

Round 1
Reviewer 1 Report
This is a comprehensive literature review on the metabolites of the respiratory epithelial lining fluid of patients with COPD showing an association with different types of COPD vs healthy controls.
Author Response
Authors' response: Thank you for your time and the revision of our manuscript.
Reviewer 2 Report
The manuscript was well written with acceptable language and layout. However, for a review article, the contents are a bit too diverse and confused. I suggest that you have to recapitulate the methods of analysis for identifications of different metabolomics phenotypes, and provide the latest suggestions of their treatment. Are there any difference in their prognosis ?
Maybe you can suggest a future research direction in terms of diagnosis, treatment or prevention.
A flow chart summarizing the above information will be fine and easy to read.
(1) This is a review article and should be more conclusive. The authors should give some opinion about the future direction of the related studies. (2) The topic is fine and worthy to be discussed. (3) It is a review article, there is nothing added. (4) It would be much better if the authors summarize the diagnosis, treatment and prognosis of every phenotype in a table form. (5) The references are fine (6) The tables and figures are fine.Author Response
We thank you for your insightful review of our manuscript and for the opportunity to revise it. Below, we enclose our responses to your specific comments and information on the major changes made to the original version of the manuscript. All these revisions were marked in red in the manuscript file. We hope that these revisions are consistent with the intentions of the Reviewer and will increase the chances for publication of the manuscript in Cells. Answers for the Reviewer are attached in a Word file.

Reviewer 3 Report
The authors present a systematic review of the respiratory metabolome in COPD patients in relation to clinical features, functional tests, and imaging studies ant they want to see if the respiratory metabolomic can distinguish between clinical phenotypes of COPD. The topic of the article is relatively novel, and there are still few review studies in this area. The article is reasonably well-organized and well-written, providing a summary and generalization of previous work. However, the paper's conclusions are not yet satisfactory and should be explained in greater detail. The authors need to address and elaborate on the following points.
1. There are now 14 studies included in the review, which is a relatively small sample size when compared to an usual systematic review and makes it challenging to draw conclusions that are significant. Right now, it seems like the search has been going on for too long. If the authors included a few more relevant papers from 2021 and 2022 (PMID: 35159156; PMID: 34801592, etc.), the sample size may be increased.
2. Aside from the small sample size, the authors did not discuss the paper's limitations. As a systematic review, the current study's limitations and shortcomings should be adequately addressed so that the reader of the paper is convinced of the findings and has a prior idea of the study's application and future direction.
3. “None of the biomarkers in the analyzed articles was identified more than once despite similar study designs.” The discussion and analysis addressing this issue is clearly not convincing enough for the reader.
4. Many of the studies in the selected sample chose COPD patients and non-smoking populations and concluded that the compounds that distinguished COPD patients from non-COPD patients belonged to the group of lipids, VOCs and amino acids. However, the objective of this paper is to assess whether the respiratory metabolome allows differentiation between clinical phenotypes of COPD (such as emphysema, eosinophilic phenotype and frequent exacerbations). No relevant research data were presented that would support the objectives of the study.
Author Response
We thank you for your insightful review of our manuscript and for the opportunity to revise the manuscript. We very much appreciate your voluntary effort undertaken to improve the manuscript quality. Your comments and suggestions were inspiring and immensely useful during the process of in-depth manuscript revision. Below, we enclose our responses to your specific comments and information on the major changes made to the original version of the manuscript. Besides, minor changes and reformulations were made to improve the manuscript's quality. All these revisions were marked in red in the manuscript file. We hope that these revisions are consistent with the intentions of the Reviewers and will increase the chances for publication of the manuscript in Cells. Answers for the Reviewer are attached in a Word file.

Round 2
Reviewer 2 Report
You have already revised the manuscript according to my comments.
Reviewer 3 Report
I have no further suggestions; the paper can now be published in its current form.